# Responding to COVID-19: The Suitability of Primary Care Infrastructure in 33 Countries

**DOI:** 10.3390/ijerph192417015

**Published:** 2022-12-18

**Authors:** Adam Windak, Katarzyna Nessler, Esther Van Poel, Claire Collins, Ewa Wójtowicz, Liubove Murauskiene, Kathryn Hoffmann, Sara Willems

**Affiliations:** 1Department of Family Medicine, Jagiellonian University Medical College, 31-061 Krakow, Poland; 2Department of Public Health and Primary Care, Ghent University, 9000 Ghent, Belgium; 3Research Centre, Irish College of General Practitioners, D02 XR68 Dublin, Ireland; 4Public Health Department, Faculty of Medicine, Vilnius University, LT-03101 Vilnius, Lithuania; 5Unit Health Services Research and Telemedicine in Primary Care, Department of Preventive- and Social Medicine, Center for Public Health, Medical University of Vienna, Kinderspitalgasse 15, 1090 Wien, Austria

**Keywords:** primary care/family medicine, general practice, infrastructure, COVID-19 pandemic, pandemic preparedness, quality of care, infection prevention and control, patient safety, PRICOV-19

## Abstract

COVID-19 proved that primary care (PC) providers have an important role in managing health emergencies, such as epidemics. Little is known about the preparedness of primary care practice infrastructure to continue providing high quality care during this crisis. The aim of this paper is to describe the perceived limitations to the infrastructure of PC practices during COVID-19 and to determine the factors associated with a higher likelihood of infrastructural barriers in providing high quality care. This paper presents the results of an online survey conducted between November 2020 and November 2021 as a part of PRICOV-19 study. Data from 4974 practices in 33 countries regarding perceived limitations and intentions to make future adjustments to practice infrastructure as a result of the COVID-19 pandemic were collected. Approximately 58% of practices experienced limitations to the building or other practice infrastructure to provide high-quality and safe care during the COVID-19 pandemic, and in 54% making adjustments to the building or the infrastructure was considered. Large variations between the countries were found. The results show that infrastructure constraints were directly proportional to the size of the practice. Better pandemic infection control equipment, governmental support, and a fee-for-service payment system were found to be associated with a lower perceived need for infrastructural changes. The results of the study indicate the need for systematic support for the development of practice infrastructure in order to provide high-quality, safe primary care in the event of future crises similar to the COVID-19 pandemic.

## 1. Introduction

Primary health care (PHC) is often described as the base of a strong health care system and the 2030 Agenda for Sustainable Development [1,2]. It has been demonstrated across a wide range of international settings that greater investment in primary health care is associated with improved population health outcomes, reduced secondary care usage, and lower overall healthcare costs [3,4,5].

The rapid spread of the COVID-19 epidemic led to the declaration of a global pandemic by the World Health Organization (WHO) on 11 March 2020 [6]. This pandemic has resulted in an unprecedented challenge for the whole healthcare sector. Moreover, it has been a major test for primary care (PC) systems. 

Shortages of the workforce, financial resources, and equipment were visible in European primary care even before the crisis caused by the COVID-19 pandemic [7,8]. Moreover, while the healthcare systems of most European countries have prioritized hospital-centered management of non-communicable diseases, the capacity to prevent, control, and treat infectious diseases has not been given adequate attention in the community and on a public health level [9].

PC providers are at the forefront of community healthcare and, as COVID-19 undoubtedly proved, have an important role in managing health emergencies, such as infectious disease epidemics. PC practices have undergone various structural and organizational changes to continue providing the quality care their communities require. Therefore, it is important to use lessons learnt from the COVID-19 pandemic to develop strong, sustainable, and resilient primary care systems.

An increasing number of papers are being published which analyze how a lack of pandemic preparedness and other challenges mentioned above have been dealt with within European countries and what are the possible directions for a transformation of the primary care system in Europe [10,11,12,13,14,15,16]. However, little is known about the preparedness of primary care practices in terms of their infrastructure.

A recently published article indicates physical infrastructure as one of the key components of any healthcare facility [17]. The authors underline that to improve the accessibility, availability, and quality of health care services, it is essential to detect and eliminate infrastructural deficiencies. 

The aim of this paper is to describe the perceived limitations to the building and infrastructure of primary care practices in times of COVID-19 as well as the infrastructural changes planned as a consequence of the pandemic. Furthermore, this analysis aims to determine the factors associated with a higher likelihood of perceived infrastructural barriers. 

## 2. Materials and Methods

### 2.1. Study Design

The PRICOV-19 project has been described in detail elsewhere [18]. Briefly, PRICOV-19 was conducted by an international consortium formed by 45 institutions from 38 countries and coordinated by Ghent University (Belgium). The ambition of the project was to explore the impact of COVID-19 on the quality and safety of care provided in primary care.

For this purpose, based on the review of the literature and experts’ experience, a 53-item questionnaire was developed to explore the following seven areas: (a) Infection prevention; (b) patient flow for COVID and non-COVID care; (c) dealing with new knowledge and protocols; (d) communication with patients; (e) collaboration; (f) well-being of the staff; and (g) characteristics of the respondent and the practice. The questionnaire was piloted, rigorously translated into the national languages of the consortium partners and uploaded into the Research Electronic Data Capture (REDCap) platform to collect data from the respondents [19].

### 2.2. Sampling and Study Participants

The data reported here were collected from 33 countries between November 2020 and November 2021, except for Belgium, where data were partially collected earlier (Figure 1 presents the analyzed countries). The data collection period varied between countries from 3 to 35 weeks. 

In each country, the consortium partners recruited the PC practices following a pre-established recruitment procedure. The study protocol assumed the recruitment of 80 to 200 practices, proportionally to the size of the country’s population. Drawing a randomized sample among all registered PC practices in the country was preferred over convenience sampling. At least one reminder was sent in all countries. A random sample of practices was undertaken in some countries. A mixed sample was drawn in other countries, adding a random sample to a convenience sample when the first did not reach enough participants.

One questionnaire was completed per practice, preferably by a general practitioner (GP)/family practitioner or by a staff member familiar with the practice organization. The overall response rate was 27.8%. However, the response rates varied between countries, and, generally, targeted convenience samples attracted larger response rates. Detailed information regarding the study design and the respondents was described elsewhere [18].

### 2.3. Outcome Variables

Two of the survey questions were selected as the outcome variables: (1) Since the COVID-19 pandemic, did you experience any limitations related to the building or the infrastructure of this practice to provide high-quality and safe care? and (2) Did the COVID-19 pandemic lead this practice to consider making adjustments in the future to the building or the infrastructure? The original answer options were: To a large extent, to a limited extent, hardly, none, and I do not know. Cases missing one of the two questions were excluded from the analyses.

### 2.4. Explanatory Variables

Following initial exploration, the following variables were included as explanatory variables: Practice characteristics, infection control infrastructure equipment, efforts to safeguard the well-being of the staff members by the practice, and good practice organization following government guidelines.

#### 2.4.1. Practice Characteristics 

Four practice characteristics were selected: Practice size, location, payment system, multidisciplinarity, and presence of a GP trainee in the practice team. To measure practice size, the number of GPs in the practice was used. The practice location was determined as big (inner) city, suburb, small town, mixed urban/rural, and rural. Five categories regarding the payment model were determined: Capitation, fee-for-service, salary, mix of salary and other, and other payment systems. A multidisciplinary team was defined as having at least one other clinical/paramedical discipline working in practice apart from a GP (the disciplines were listed in the questionnaire: Nurse or nurse assistant; dietician or nutritionist; health promotor; physiotherapist, manual therapist, osteopath; podologist; psychologist). The respondents indicated if the above professionals worked in the practice or not. The total number of disciplines ticked was represented by this variable. Due to the data distribution for the number of GP trainees (no GP trainees in more than half of the practices), a dichotomous variable was created (practices with and without GP trainees).

#### 2.4.2. Infection Control Infrastructure Equipment

A seven-item infection control infrastructure equipment (ICIE) score was created based on having each of the seven infection control items in every consulting room (sink, non-contact tap, non-contact bin, disposable gloves, disposable coats, surface disinfectant, and paper cover for examination table) [20]. For this paper, practices which scored the number of items from 0 to 4 were grouped into one category. The “ideal” practices were considered the ones scoring a total of seven items.

#### 2.4.3. Safeguarding the Well-Being of the Staff Members by the Practice

There were nine strategies listed on the questionnaire and practices indicated which they had implemented (Appendix A). If implemented, each was given one point and a total score was created ranging from 0–9.

#### 2.4.4. Adequate Government Support for the Proper Functioning of Practice

The sense of obtaining adequate government support for the proper functioning of the practice during the pandemic was assessed on the basis of answers to a question with a five-point Likert scale. The analysis was conducted with “strongly agree” as the most positive category taken as the reference.

### 2.5. Statistical Analysis

Statistical analysis was performed by SPSS software (version 28.0.1.0 SPSS Inc., Chicago, IL, USA) using version 7 of the database, which was the version consisting of the cleaned data of 33 countries available as of 3 November 2021. Ghent University was responsible for data cleaning. 

Due to the clustering of respondent practices in countries, we ran mixed models using logistic regression. The experienced limitations to the practice’s infrastructure and the considered changes to the practice were included in the models as the dichotomous outcome variables after grouping answers (“none” and “hardly” versus “to a limited extent” and “to a large extent”). For each of the two outcomes, four models were tested using a stepwise approach with the null model (Model I) permitting the calculation of the intraclass correlation coefficient (ICC), assessing the proportion of the variance in the outcome variable that can be explained by country. In subsequent models, we added practice characteristics (Model II) and COVID-19 context characteristics (Model III and Model IV) as fixed effects. The Akaike’s Information Criterion (AIC) and −2 log likelihood values were used as goodness-of-fit model criteria. The likelihood ratio test was used to compare model fit between the nested models. The boundary values for the criterion of statistical significance (p, two-fold) were determined at *p* < 0.05.

### 2.6. Ethical Approval

The study was conducted in accordance with the Helsinki Declaration and was approved by The Research Ethics Committee of Ghent University Hospital (BC-07617). Moreover, when required by national regulations, research ethics committees in the different partner countries provided additional approval. All participants provided online informed consent prior to answering the questions.

## 3. Results

### 3.1. Practice Characteristics

The complete answers to the questions related to the infrastructural constraints during the COVID-19 pandemic were provided by 4974 practices from 33 countries participating in PRICOV-19 study. Almost one-third were in the big cities, 18% in the villages, while the rest were in suburbs or mixed urban/rural settings. Overall, 40% of the practices’ medical services were reimbursed on a fee-for-service basis, idem for capitation, while the payment in the rest was based on other/mixed methods. One-third of the respondents represented single practices and almost one-fourth employed six or more GPs. GP trainees were present in 44% of practices. One-third of the practices were fully equipped to support infection control with all seven infrastructure items in each consulting room. Detailed information on the practice characteristics is presented in Table 1.

#### 3.1.1. Perceived Governmental Support

Less than a quarter of the respondents agreed that the government provided adequate support for the functioning of primary care practices during the pandemic, a similar proportion had a neutral opinion about it, while the rest disagreed or strongly disagreed with this opinion. The details are shown in Table 2.

#### 3.1.2. Safeguarding the Well-Being of the Staff Members by the Practice

Several strategies were implemented by the practices to secure the well-being of the staff during the COVID-19 pandemic; limiting the number of patients in the waiting room, telephone triage, and increasing infection control measurements were the most common. 

The exact data are presented in the Appendix A. 

### 3.2. Perceived Limitations and Needs for Changes in Infrastructure

Almost two-third of the respondents (58%) experienced limitations (to a large or limited extent) to the building or other practice infrastructure to provide high-quality and safe care during the COVID-19 pandemic. Over half of respondents (54%) reported that they have considered making adjustments (to a large or limited extent) to the building or the infrastructure (Table 3). Large variations between the countries were found (Figure 1 and Figure 2). No significant regional differences were found; however, we observed the trend in which the respondents from Western European countries reported least frequently both experiencing limitations and considering making adjustments in the future. Perceived limitations to the practice infrastructure and having considered making adjustments in the future to the building or the infrastructure of the practice were strongly (Gamma correlation 0.60) and significantly (*p* = 0.00) correlated (Figure 3).

### 3.3. Correlation with Practice Characteristics

Table 4 presents the results of the logistic mixed model analysis of the outcome variables and the practice characteristics. Model I, showing the null model or intercept-only model, has an ICC = 15%, meaning that 15% of risk/chances of experiencing limitations is explained by between-country differences. Each subsequent stepwise model shows a better goodness-of-fit (based on smaller AIC and –2 log likelihood values). The likelihood ratio test shows that each model fits significantly better than the previous one and the variances reduce when adding predictors. Three of five practice characteristics are independently and significantly related to the experience of limitations in infrastructure; its size is expressed by the number of GPs, payment system, and GP trainees present in the practice. Compared with GPs working in single-handed practices, GPs working in larger practices experienced more infrastructure constraints: The higher the number of GPs, the more likely the practice was to report infrastructural limitations. The practices paid on a fee-for-service basis had a lower likelihood to report limitations compared to those paid by capitation. Practices with GP trainees had a higher likelihood of limitations than those without.

Better safeguarding of the well-being of the staff members by the practice resulted in lower levels of perceived limitations. Practices in which not all consultation rooms were equipped with the seven specified items experienced significantly more constraints in the infrastructure in general (both Models III and IV confirmed a negative trend). Experiencing adequate governmental support was a protective factor.

Table 5 presents the results of the logistic mixed model analysis of the considered future adjustments to the building or the practice infrastructure and practice characteristics. Model I, showing the null model or intercept-only model, has an ICC = 12%, meaning that 12% of the risk/chances of considering adjustments to the building or the infrastructure is attributable to the country level. Each subsequent stepwise model shows a better goodness-of-fit (based on smaller AIC and –2 log likelihood values). The likelihood ratio test showed that each model fits significantly better than the previous one. All five practice characteristics were independently and significantly related to considering future adjustments to the building or practice infrastructure. The need for changes was reported more frequently in the practices located in rural areas, staffed by a higher number of GPs and other professionals as well as those employing GP trainees. The significance of these four characteristics was confirmed in all models, except in Models III and IV for practices employing 4–5 GPs. The practices paid on a fee-for-service basis considered less frequently infrastructural changes then those with a capitation payment system. However, the significance of this association was confirmed only in Models III and IV. Better safeguarding of the well-being of the staff members increased the risk of infrastructural changes being considered by the respondents. The role of adequate governmental support had no significant association with considering infrastructural changes.

## 4. Discussion

### 4.1. Summary of Main Findings 

Perceived infrastructure constraints were directly proportional to the size of the practice, expressed by the number of physicians. The limitations were more likely to be perceived in practices conducting postgraduate training in family medicine. Better pandemic infection control equipment was associated with less perceived need for infrastructure changes during the pandemic. The results of this study indicate the important role of payment systems for primary care. The practices paid on a fee-for-service basis were less likely to report a perceived need for infrastructural changes than practices funded by other means. Moreover, the study shows the importance of governmental support. In practices where this support was considered to be sufficient, the need for future changes to the infrastructure was less often reported. The experience of limitations and planned future changes in infrastructure was reported least frequently by Western European respondents.

### 4.2. Comparison with Existing Literature

The pandemic has led to a greater awareness of deficiencies and to the reorganization of primary care [21,22]. This experience is also why practices in our study report the need for infrastructural constraints in their practices.

Our results show that the payment system within the practices impacts the perception of infrastructural constraints. This has been noted elsewhere where the payment system impacts on investment, service delivery, and behavior [23]. Previous work has shown a higher safety culture in training compared to non-training practices, which is supported by our findings, as well [24]. Moreover, it is propounded that practices involved in education consider themselves as role-models and are therefore more aware of the need for constant improvement [25]. Practices in our study report the need to improve their infrastructure to provide for patients during and post-pandemic. Improvements to ensure safe care regarding primary care facilities is also called for in terms of overall infection control measures [20]. Furthermore, practices consider that government support was insufficient and increased governmental support for primary care has been called for elsewhere [26,27].

### 4.3. Strengths and Limitations

To the best of our knowledge, PRICOV-19 is the largest and most comprehensive study on the organization of PC practices during COVID-19. This resulted in a large, sample of practices surveyed across 33 countries. Moreover, this study has the strength that the used survey was developed and validated in several phases, including a pilot study in Flanders. 

However, several limitations should also be noted. One limitation is that our survey was based on a self-selecting sample, which comes with inherent bias. Second, the sample was obtained in different ways depending on the country. Third, the data collection was carried out in a period of 13 months, and therefore covered different phases of the pandemic. Although the survey design provides a broad overview of the issues raised, it cannot provide a specific explanation of the answers provided, the patterns observed, or reasons for changes over time.

### 4.4. Implications for Research, Policy, and Practice

Infection lulls permitted a significant investigation on the suitability of hospital structures recently challenged by unforeseen infections [28]. Our research demonstrates that it would also be beneficial to investigate the limitations and opportunities for improving infrastructure on the front lines of health care, given that the majority of respondents reported limitations linked to practice infrastructure. 

The study indicated that 50% of respondents believed that government assistance for the operating of the practices was insufficient. Other research has shown that rather than short-term adaptations that cover flaws, long-term adaptations that improve practices could be of considerable value [29]. Investment in buildings and the increased supply of infection control tools require increased funding and payment arrangements, which should be examined considering the necessity for adequate financing.

Our findings showed that the availability of infection control equipment decreased the perceived need for infrastructure improvements, thus allowing for less disruption when coping with unforeseen shocks. 

The results can be used for practical and research-based institutional and capacity planning, for developing primary care infrastructure.

## 5. Conclusions

The results of the study indicate the need for systematic support for the development of practice infrastructure in order to provide high-quality, safe primary care in the event of future crises similar to the COVID-19 pandemic.

## Figures and Tables

**Figure 1 ijerph-19-17015-f001:**
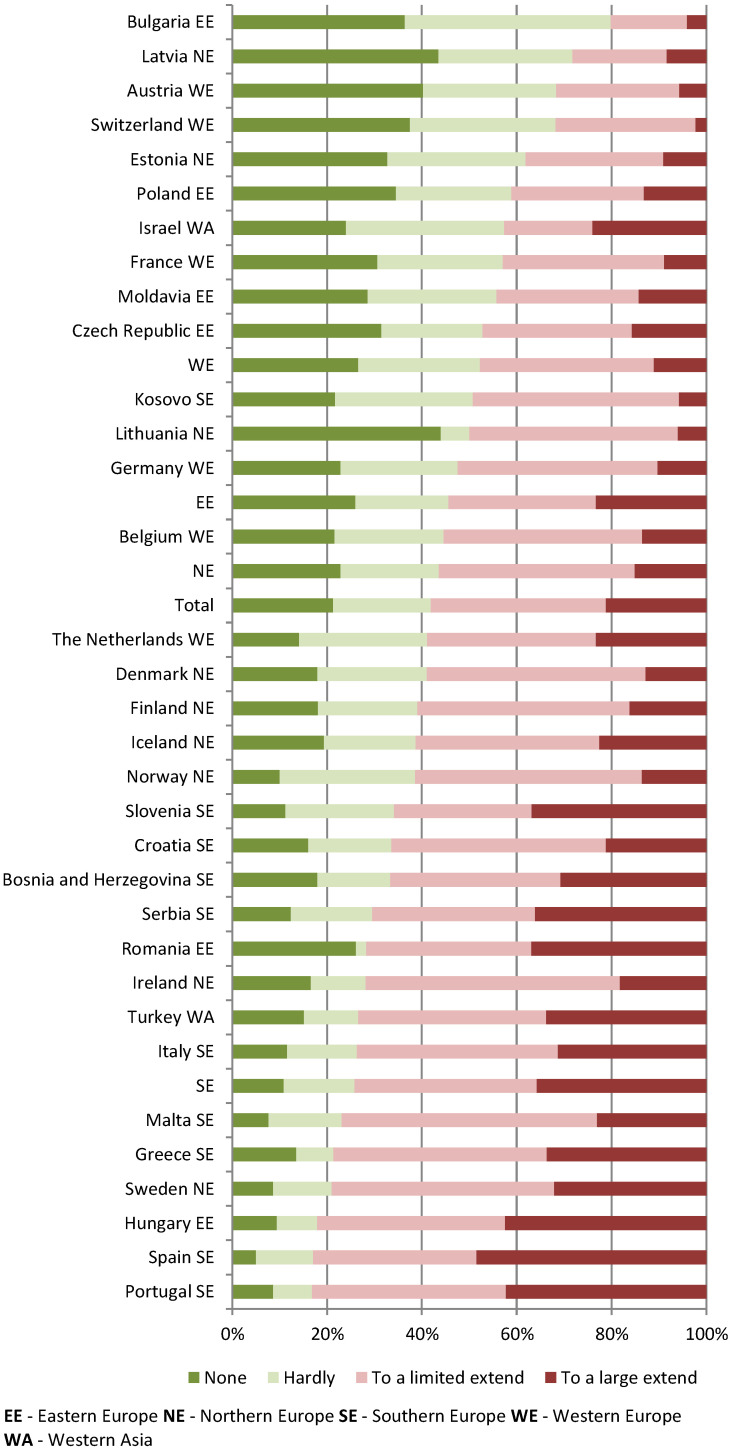
Experience of limitations related to the building or practice infrastructure.

**Figure 2 ijerph-19-17015-f002:**
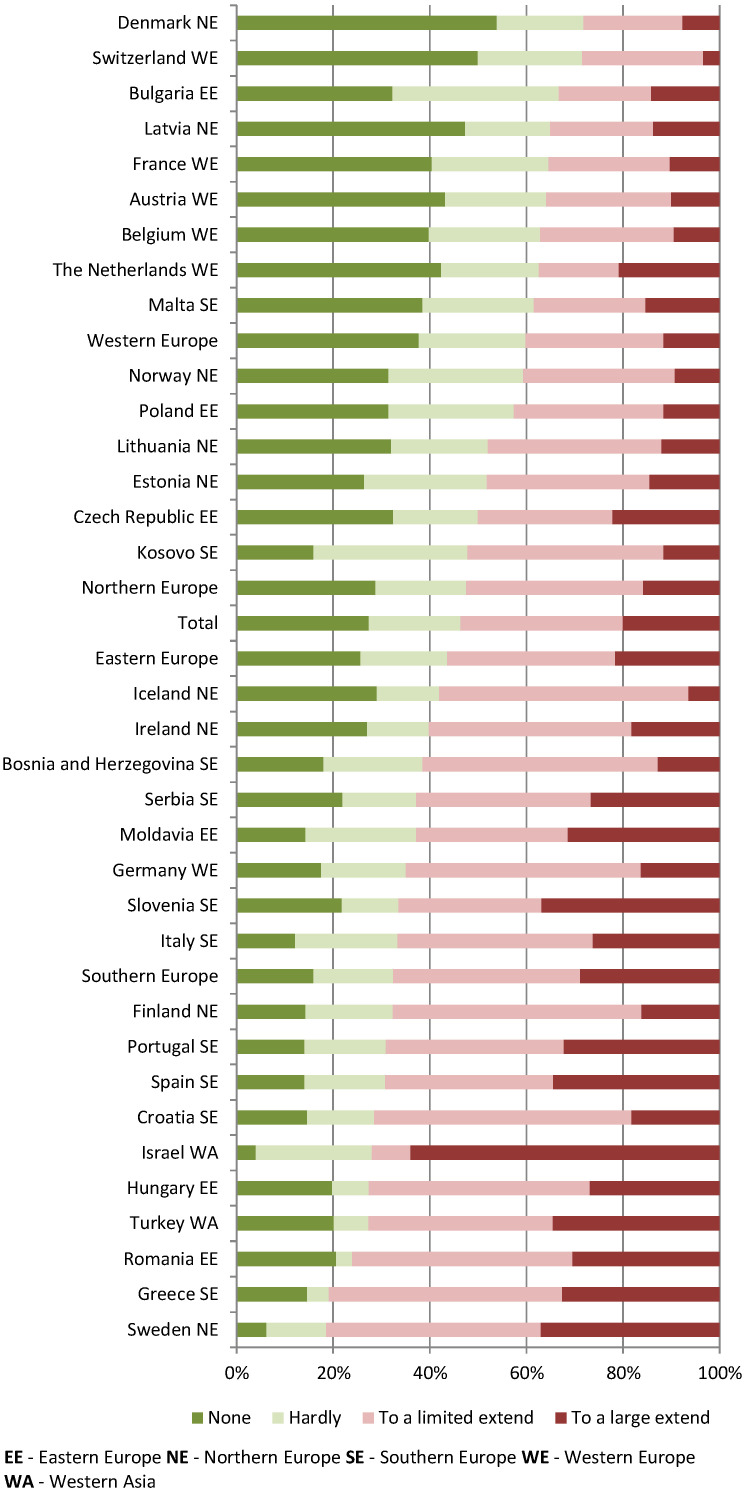
Considering making adjustments in the future to the building or the infrastructure.

**Figure 3 ijerph-19-17015-f003:**
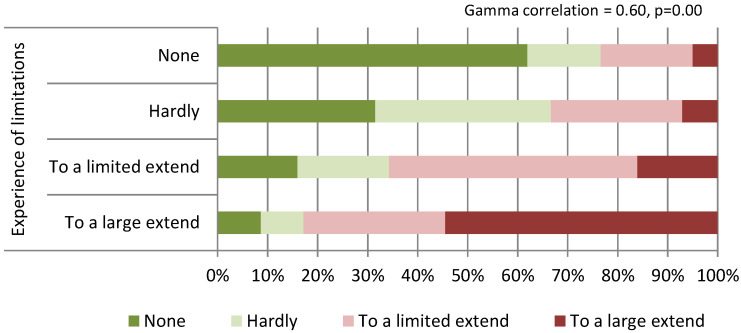
Correlation of experiencing limitations and considering making adjustments.

**Table 1 ijerph-19-17015-t001:** Main characteristics of the practices participating in the study.

Total	N 4974	% 100
**Location of practice**		
Big (inner) city	1613	32.4
Suburbs	514	10.3
(Small) town	923	18.6
Mixed urban-rural	1013	20.4
Rural Missing value	898 13	18.1 0.3
**Payment system**		
Capitation	2002	40.9
Fee-for-service	1957	40.0
Salary+ mix of salary and other	507	10.4
Other Missing value	426 82	8.7 1.6
**Number of GPs**		
1	1640	33.0
2-3	1320	26.5
4-5	786	15.8
6+ Missing value	1167 61	23.5 1.2
**GP trainees**		
No	2758	55.4
Yes Missing value	2178 38	43.8 0.8
**Number of clinical professions** *		
1	1669	33.6
2	2151	43.2
3+ Missing value	1154 0	23.2 0
**Infection equipment indicator**		
0-4 items	382	7.7
5 items	970	19.5
6 items	1627	32.7
7 items Missing value	1495 500	30.1 10.1

* (1) GP, (2) nurse or nurse assistant, (3) dietician or nutricionist, (4) health promotor, (5) physiotherapist, manual therapist, osteopath, (6) podologist, (7) psychologist.

**Table 2 ijerph-19-17015-t002:** Respondents’ opinion on government’s role on the functioning of practice during the COVID-19 pandemic.

	Support for Proper Functioning	
Total	N 4974	% 100
Strongly disagree	823	16.5
Disagree	1419	28.5
Neutral	1014	20.4
Agree	830	16.7
Strongly agree	189	3.8
Missing value	699	14.1

**Table 3 ijerph-19-17015-t003:** Experienced limitations to the practice infrastructure and considered making adjustments to the infrastructure as an effect of the COVID-19 pandemic.

	Experienced Limitations		Considered Making Adjustments	
	N 4974	% 100	N 4974	% 100
None	1056	21.2	1362	27.4
Hardly	1028	20.7	942	18.9
To a limited extent	1835	36.9	1674	33.7
To a large extent	1055	21.2	996	20.0

**Table 4 ijerph-19-17015-t004:** Results of mixed effects logistic regression analysis of potential predictors for an experience of limitations related to the building or the practice infrastructure.

	Model I	Model II *p* < 0.001	Model III *p* < 0.001	Model IV *p* < 0.001
	OR (95% CI)	OR (95% CI)	OR (95% CI)	OR (95% CI)
**Intercept**	1.43 (1.10; 1.87) *p* = 0.008	1.10 (0.80; 1.51)	0.69 (0.47; 1.03)	0.42 (0.26; 0.69) *p* < 0.001
**Location of practice**		(*p* = 0.673)	(*p* = 0.455)	(*p* = 0.607)
Big (inner) city		Ref.	Ref.	Ref.
Suburbs		0.90 (0.72; 1.12)	0.93 (0.73; 1.18)	0.94 (0.74; 1.20)
(Small) town		0.99 (0.82; 1.19)	1.06 (0.86; 1.29)	1.04 (0.85; 1.27)
Mixed urban-rural		1.07 (0.90; 1.29)	1.15 (0.95; 1.39)	1.13 (0.93; 1.37)
Rural		1.02 (0.84; 1.23)	1.09 (0.89; 1.34)	1.09 (0.88; 1.34)
**Payment system**		(*p* = 0.019)	(*p* = 0.006)	(*p* = 0.008)
Capitation		Ref.	Ref.	Ref.
Fee-for-service		0.78 (0.62; 0.98) *p* = 0.037	0.74 (0.58; 0.94) *p* = 0.014	0.73 (0.57; 0.94) *p* = 0.014
Salary+ mix of salary and other		1.28 (0.68; 2.43)	1.45 (0.75; 2.78)	1.37 (0.71; 2.63)
Other		1.27 (0.92; 1.75)	1.27 (0.91; 1.78)	1.26 (0.90; 1.77)
**Number of GPs**		(*p* < 0.001)	(*p* < 0.001)	(*p* < 0.001)
1		Ref.	Ref.	Ref.
2–3		1.57 (1.31; 1.87) *p* < 0.001	1.54 (1.28; 1.85) *p* < 0.001	1.53 (1.27; 1.85) *p* < 0.001
4–5		1.46 (1.18; 1.81) *p* < 0.001	1.38 (1.09; 1.74) *p* = 0.006	1.39 (1.10; 1.76) *p* = 0.005
6+		1.88 (1.49; 2.39) *p* < 0.001	1.77 (1.38; 2.28) *p* < 0.001	1.86 (1.44; 2.40) *p* < 0.001
**Clinical professions (1–7)/multidisciplinary team**		0.96 (0.90; 1.02) *p* = 0.203	0.96 (0.90; 1.03) *p* = 0.231	0.94 (0.87; 1.02) *p* = 0.141
**GP trainees**				
No		Ref.	Ref.	Ref.
Yes		1.16 (1.01; 1.34) *p* = 0.035	1.16 (0.997; 1.34) *p* = 0.055	1.19 (1.02; 1.38) *p* = 0.027
**Infection equipment indicator**			*p* = 0.055	*p* = 0.107
7 items			Ref.	Ref.
6 items			1.10 (0.94; 1.30)	1.08 (0.92; 1.27)
5 items			1.20 (0.999; 1.45) *p* = 0.051	1.17 (0.97; 1.42)
0–4 items			1.41 (1.07; 1.85) *p* = 0.013	1.38 (1.05; 1.83) *p* = 0.023
**Safeguarding the well-being (score 0–9)**			1.06 (1.02; 1.11) *p* = 0.002	1.07 (1.02; 1.11) *p* = 0.002
**Adequate government support**				*p* < 0.001
Strongly agree				Ref.
Agree				1.23 (0.87; 1.73)
Neutral				1.30 (0.92; 1.83)
Disagree				1.88 (1.33; 2.64) *p* < 0.001
Strongly disagree				2.15 (1.50; 3.07) *p* < 0.001
Intercept variance (s.e.)	0.56 (0.15) *p* < 0.001	0.44 (0.13) *p* < 0.001	0.40 (0.12) *p* < 0.001	0.35 (0.10 *p* < 0.001
Model information				
Akaike’s Information Criterion (AIC)	21,827.13	21,178.64	18,596.34	18,277.96
−2 Log Likelihood	21,825.13	21,176.64	18,594.34	18,275.96
Likelihood ratio test		652.49 (df = 17) *p* < 0.001	2582.30 (df = 12) *p* < 0.001	318.38 (df = 4) *p* < 0.001

**Table 5 ijerph-19-17015-t005:** Results of mixed effects logistic regression analysis of potential predictors for considering making adjustments to the building or the infrastructure.

	Model I	Model II (*p* < 0.000)	Model III (*p* = 0.000)	Model IV (*p* = 0.000)
	OR (95% CI)	OR (95% CI)	OR (95% CI)	OR (95% CI)
**Intercept**	1.28 (1.01; 1.62) *p* = 0.044	0.76 (0.56; 1.03) *p* = 0.080	0.36 (0.24; 0.53) *p* < 0.001	0.32 (0.19; 0.53) *p* < 0.001
**Location of practice**		(*p* = 0.063)	(*p* = 0.017)	(*p* = 0.024)
Big (inner) city		Ref.	Ref.	Ref.
Suburbs		0.98 (0.78; 1.22)	0.94 (0.75; 1.20)	0.95 (0.75; 1.21)
(Small) town		0.93 (0.77; 1.11)	0.92 (0.76; 1.12)	0.90 (0.74; 1.10)
Mixed urban-rural		1.14 (0.95; 1.36)	1.16 (0.96; 1.41)	1.14 (0.94; 1.38)
Rural		1.21 (1.002; 1.46) *p* = 0.048	1.28 (1.04; 1.56) *p* = 0.019	1.25 (1.02; 1.54) *p* = 0.031
**Payment system**		(*p* = 0.204)	(*p* = 0.093)	(*p* = 0.043)
Capitation		Ref.	Ref.	Ref.
Fee-for-service		0.83 (0.66; 1.04)	0.79 (0.62; 1.01) *p* = 0.063	0.75 (0.59; 0.97) *p* = 0.025
Salary+ mix of salary and other		0.87 (0.49; 1.57)	0.76 (0.40; 1.44)	0.71 (0.37; 1.36)
Other		1.16 (0.85; 1.59)	1.22 (0.87; 1.70)	1.20 (0.86; 1.69)
**Number of GPs**		(*p* = 0.008)	(*p* = 0.027)	(*p* = 0.023)
1		Ref.	Ref.	Ref.
2–3		1.27 (1.07; 1.52) *p* = 0.007	1.30 (1.08; 1.57) *p* = 0.006	1.31 (1.09; 1.59) *p* = 0.005
4–5		1.27 (1.02; 1.57) *p* = 0.033	1.18 (0.94; 1.49	1.21 (0.96; 1.53)
6+		1.47 (1.16; 1.85) *p* = 0.001	1.36 (1.05; 1.74) *p* = 0.018	1.38 (1.07; 1.77) *p* = 0.013
**Clinical_professions (1–7)/multidisciplinary team**		1.11 (1.04; 1.18) *p* < 0.001	1.08 (1.01; 1.16) *p* = 0.023	1.10 (1.01; 1.19) *p* = 0.022
**GP trainees**				
No		Ref.	Ref.	Ref.
Yes		1.27 (1.11; 1.46) *p* < 0.001	1.25 (1.08; 1.45) *p* = 0.003	1.27 (1.09; 1.48) *p* = 0.002
**Infection equipment indicator**			(*p* = 0.005)	(*p* = 0.006)
7 items			Ref.	Ref.
6 items			1.07 (0.92; 1.26)	1.06 (0.90; 1.25)
5 items			1.30 (1.08; 1.57) *p* = 0.006	1.30 (1.07; 1.57) *p* = 0.007
0–4 items			0.85 (0.65; 1.10)	0.84 (0.64; 1.10)
**Safeguarding the well-being (score 0–9)**			1.16 (1.11; 1.20) *p* < 0.001	1.16 (1.12; 1.21) *p* < 0.001
**Adequate government support**				(*p* < 0.001)
Strongly agree				Ref.
Agree				1.08 (0.76; 1.53)
Neutral				1.05 (0.74; 1.48)
Disagree				1.46 (1.04; 2.06) *p* = 0.029
Strongly disagree				1.51 (1.05; 2.16) *p* = 0.024
Intercept variance (s.e.)	0.44 (0.12) *p* < 0.001	0.41 (0.12) *p* < 0.001	0.42 (0.12) *p* < 0.001	0.40 (0.12) *p* < 0.001
Model Information				
AIC	21,588.44	20,952.08	18,480.54	18,149.25
−2 Log Likelihood	21,586.44	20,950.08	18,478.54	18,147.25
Likelihood ratio test		636.36 (df = 17) *p* < 0.001	2471.54 (df = 12) *p* < 0.001	331.29 (df = 4) *p* < 0.001

## Data Availability

The anonymized data are held at Ghent University and are available to participating partners for further analysis upon signing an appropriate usage agreement.

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
