# Peer review of "Responding to COVID-19: The Suitability of Primary Care Infrastructure in 33 Countries"

_ijerph, 2022, doi:10.3390/ijerph192417015_

Round 1

Reviewer 1 Report

1) Unfortunately, there is inconsistency in how the terms “primary health care (PHC)” and “primary care” are used. THESE ARE NOT INTEREXCHANGABLE TERMS – but mean different things.

In some instances (the Title, Lines 63, 69, 187, 284, 317) they are reflected correctly, while in others (Lines 18, 21, 46, 53, 55, 58, 62, 92, 94, I could go on but I think you get the point) they are incorrectly used. This list is not exhaustive, so please look at all occurrences.

To be clear - “PHC” is an approach (a concept not unlike UHC), while “primary care” is the delivery of services (among other things). Hence there are no “PHC facilities” but “primary care facilities”. There are no “PHC providers” but “primary care providers”, or “PHC systems”, but there are "primary care systems". Just as there are not “UHC facilities” or “UHC providers”.

I believe it is critical for us to use the terms consistently if we ever hope to have the global health world use them accurately.

2) The literature and references researched is commendable.

 3) The sample size of survey respondents is impressive.

 4) The rationale of why the number of GPs was used as opposed to patient load to measure practice size would be insightful.

Author Response

Response to Reviewer 1 Comments

Point 1:

Unfortunately, there is inconsistency in how the terms “primary health care (PHC)” and “primary care” are used. THESE ARE NOT INTEREXCHANGABLE TERMS – but mean different things.

In some instances (the Title, Lines 63, 69, 187, 284, 317) they are reflected correctly, while in others (Lines 18, 21, 46, 53, 55, 58, 62, 92, 94, I could go on but I think you get the point) they are incorrectly used. This list is not exhaustive, so please look at all occurrences.

To be clear - “PHC” is an approach (a concept not unlike UHC), while “primary care” is the delivery of services (among other things). Hence there are no “PHC facilities” but “primary care facilities”. There are no “PHC providers” but “primary care providers”, or “PHC systems”, but there are "primary care systems". Just as there are not “UHC facilities” or “UHC providers”.

I believe it is critical for us to use the terms consistently if we ever hope to have the global health world use them accurately.

Response 1:

Thank you for that comment. We have reviewed the document and made the suggested changes.

Point 2:

The literature and references researched is commendable.

Response 2:

Thank you.

Point 3:

The sample size of survey respondents is impressive.

Response 3:

Thank you.

Point 4:

The rationale of why the number of GPs was used as opposed to patient load to measure practice size would be insightful.

Response 4:

The number of GPs was closely correlated with the number of patients. Both measures could not be taken into account in regression models. As we intended to present the practice staff perspective (as they were the respondents in our study), we decided to use the number of GPs to measure practice size. It has been added to the manuscript.

Reviewer 2 Report

The paper is written very well and contributes to the understanding of the limitations of the infrastructure in primary health care both due to the COVID-19 pandemic and other obstacles. On the other hand, the work shows the necessity of changes in work that occurred to a larger or lesser extent due to the COVID-19 pandemic and also contributes to a better understanding of the factors that influenced the number of necessary changes in order for employees to better cope with the new situation. Although the entire work is based on the subjective experience of the employees of the practice itself and part of the results are expected, with a high-quality and detailed analysis of the data, we also get part of the data that will certainly be useful in future planning of resources for work, changes within the health system as well as infrastructure in primary practices.

The summary is clearly written. It would be good to add numerical data to the text when summarizing the results in order to gain credibility.

The introduction of the paper very clearly describes the need to conduct such a study (lack of existing studies on this topic, clarification of the changes in work brought about by the COVID-19 pandemic in an unprepared system and the factors that influenced them other than the pandemic itself, etc.). The objectives of the work are clearly stated and will also be well elaborated with a clear conclusion at the end.

Methods are well described with a very detailed description of the used statistical methods, which makes it easier to follow the explanation of the results. The authors state that a random sample was preferred, but in countries where the number of participants was too small in the end, the research was conducted on a convenience sample. What number of participants was considered too small? In how many countries was the survey conducted by random sampling, and in which by convenience sampling? In addition to the above, it is unclear how the authors calculated that the response rate was 27.8% (are the countries in which convenience sampling was included in that share, and what was their response rate when the study was to be conducted on a random sample?). I find this quite important for the final interpretation of the data.

Discussion – The first part of the discussion clearly and concisely presents the most important discoveries in the research. After that, the advantages and disadvantages of the conducted research are clearly presented along with a description of possible problems that arose during the collection and interpretation of data. The significance of the obtained results for practice is also evident.

References - All references except the first one are completely wrong written and do not follow the citation instructions provided on the journal website (should be corrected). The number before reference 28 is missing; after number 29, the repeated number 29 needs to be removed.

In general, the research is very important for the future of primary health care work planning, and I believe that it should be published after correcting the mentioned minor writing errors.

Author Response

Response to Reviewer 2 Comments

Point 1:

The paper is written very well and contributes to the understanding of the limitations of the infrastructure in primary health care both due to the COVID-19 pandemic and other obstacles. On the other hand, the work shows the necessity of changes in work that occurred to a larger or lesser extent due to the COVID-19 pandemic and also contributes to a better understanding of the factors that influenced the number of necessary changes in order for employees to better cope with the new situation. Although the entire work is based on the subjective experience of the employees of the practice itself and part of the results are expected, with a high-quality and detailed analysis of the data, we also get part of the data that will certainly be useful in future planning of resources for work, changes within the health system as well as infrastructure in primary practices.

The summary is clearly written. It would be good to add numerical data to the text when summarizing the results in order to gain credibility.

Response 1:

Thank you for that kind comment. We have added numerical data to the summary the document as suggested.

Point 2:

The introduction of the paper very clearly describes the need to conduct such a study (lack of existing studies on this topic, clarification of the changes in work brought about by the COVID-19 pandemic in an unprepared system and the factors that influenced them other than the pandemic itself, etc.). The objectives of the work are clearly stated and will also be well elaborated with a clear conclusion at the end.

Response 2:

Thank you for that kind comment.

Point 3:

Methods are well described with a very detailed description of the used statistical methods, which makes it easier to follow the explanation of the results. The authors state that a random sample was preferred, but in countries where the number of participants was too small in the end, the research was conducted on a convenience sample. What number of participants was considered too small? In how many countries was the survey conducted by random sampling, and in which by convenience sampling? In addition to the above, it is unclear how the authors calculated that the response rate was 27.8% (are the countries in which convenience sampling was included in that share, and what was their response rate when the study was to be conducted on a random sample?). I find this quite important for the final interpretation of the data.

Response 3:

Thank you for that comment. The details of the sampling and recruitment in the participating countries are described in Ref. 18. Van Poel E, Vanden Bussche P, Klemenc-Ketis Z, Willems S. How did general practices organize care during the COVID-19 pandemic: the protocol of the cross-sectional PRICOV-19 study in 38 countries. BMC Prim Care. 2022;23(1):11. We can read there:

“In each country, the consortium partner(s) recruited GP practices following a pre-defined recruitment procedure. Drawing a randomized sample among all GP practices in the country was preferred over convenience sampling. At least six countries were able to sample the practices randomly. A mixed sample was drawn in some countries, adding a convenience sample to the random sample when the first one did not have enough participants. In about half of the countries, a convenience sample was used. In each country, the consortium partner sent out at least one reminder. In the majority of the countries, a sample was drawn from GP practices in the entire country. Only in a couple of countries, the data collection was limited to a specific region in the country. Partners logged all the steps taken in the sampling procedure. PRICOV-19 aimed to sample between 80 and 200 GP practices per country, depending on the number of GP practices. Table 1 shows more information on the sampling and recruitment in the participating countries. The response rates were calculated by the ratio of the number of GP practices that at least filled in the first part of the questionnaire to the number of GP practices that received an invitation to participate in the study.”

We rephrased and extended information in the “Sampling and Study Participants” section of our manuscript.

Point 4:

Discussion – The first part of the discussion clearly and concisely presents the most important discoveries in the research. After that, the advantages and disadvantages of the conducted research are clearly presented along with a description of possible problems that arose during the collection and interpretation of data. The significance of the obtained results for practice is also evident.

Response 4:

Thank you for that comment.

Point 5:

References - All references except the first one are completely wrong written and do not follow the citation instructions provided on the journal website (should be corrected). The number before reference 28 is missing; after number 29, the repeated number 29 needs to be removed.

Response 5:

Thank you for that comment. The references have been corrected accordingly.

Point 6:

In general, the research is very important for the future of primary health care work planning, and I believe that it should be published after correcting the mentioned minor writing errors.

Response 6:

Thank you for that comment. We hope that the manuscript is suitable for publication after introducing the suggested changes.